Journal of Data-centric Machine Learning Research (2023)        Submitted 9/23; Revised 10/23; Published 11/23

# Deep Learning for Accurate Diagnosis of Viral Infections through scRNA-seq Analysis: A Comprehensive Benchmark Study

**Ziwei Yang**                                                    ZIWEI.YANG@UTEXAS.EDU
*Electrical and Computer Engineering*
*University of Texas at Austin*
*Austin, TX 78712, USA*

**Xuxi Chen**                                                    XXCHEN@UTEXAS.EDU
*Electrical and Computer Engineering*
*University of Texas at Austin*
*Austin, TX 78712, USA*

**Biqing Zhu**                                                    BIQING.ZHU@YALE.EDU
*Computational Biology and Bioinformatics*
*Yale University*
*New Haven, CT 06510, USA*

**Tianlong Chen**                                                    TIANLONG@CS.UNC.EDU
*Computer Science*
*University of North Carolina at Chapel Hill*
*Chapel Hill, NC 27599, USA*

**Zhangyang Wang**                                                    ATLASWANG@UTEXAS.EDU
*Electrical and Computer Engineering*
*University of Texas at Austin*
*Austin, TX 78712, USA*

**Reviewed on OpenReview:** *https: // openreview. net/ forum? id=XXXX*

**Editor:** My editor

## Abstract

Infectious disease diagnostics primarily rely on physicians' clinical expertise and rapid antigen/antibody tests, a subjective approach prone to errors due to various factors including patient history accuracy and physician experience. To address these challenges, we propose a biological evidence-based diagnostic tool using deep learning to analyze patient-derived single-cell RNA sequencing (scRNA-seq) profiles from blood samples. scRNA-seq provides high-resolution gene expression data at the single-cell level, capturing unique transcriptional signatures and immunological responses induced by different viral infections. In this work, we conducted the first-of-its-kind benchmark study to evaluate five computational models, including four deep learning-based methods (contrastiveVI, scVI, SAVER, scGPT) and PCA as a baseline - trained and evaluated on patient-derived scRNA-seq datasets carefully sourced by us. We assess their efficacy in distinguishing scRNA-seq profiles associated with various viral infections, aiming to identify distinct immunological features representative of each infection. The results demonstrate that contrastiveVI, outperforms other models in

all key performance metrics and the visual cluster performance. Furthermore, our research also underscores the substantial influence of batch effects when analyzing scRNA-seq data from multiple sources. Overall, our study successfully demonstrates that deep learning models can accurately identify the type of infection from patient plasma samples based on scRNA-seq profiles, and improve the accuracy and specificity in the diagnosis of infectious diseases. This research contributes to the development of more objective, evidence-based diagnostic methods in the infectious disease domain, potentially reducing diagnostic errors and improving patient outcomes.

# 1 Introduction

Infectious diseases continue to pose significant challenges to global health, with their diagnosis and management often relying on a combination of clinical expertise and rapid diagnostic tests Caliendo et al. (2013). Traditional approaches to infectious disease diagnostics primarily depend on physicians' subjective assessments based on patient symptoms and medical history, followed by confirmatory tests such as rapid antigen or antibody assays Croskerry et al. (2013). While this method has been the cornerstone of clinical practice, it is inherently prone to errors due to various factors Singh et al. (2017), including the accuracy of patient-reported symptoms, the comprehensiveness of medical histories, and the variability in physicians' experience.

The limitations of this subjective diagnostic approach are particularly evident in the context of emerging infectious diseases or during epidemic outbreaks, where rapid and accurate identification of pathogens is crucial for both individual patient care and public health responses. Misdiagnoses or delayed diagnoses can lead to inappropriate treatments, prolonged illness, and in severe cases, increased mortality rates. Moreover, from a public health perspective, inaccurate diagnoses can hamper efforts to control disease spread, potentially exacerbating outbreaks Balogh et al. (2015). These challenges underscore the need for more objective, evidence-based diagnostic tools that can actively identify the type of pathogen infecting a patient using a universal clinical sample.

Recent advancements in next-generation sequencing technologies, particularly single-cell RNA sequencing (scRNA-seq), have opened new avenues for understanding the molecular signatures of various diseases at unprecedented resolution. scRNA-seq provides a high-resolution view of gene expression at the single-cell level, capturing the unique transcriptional signatures and immunological responses induced by different pathogens Papalexi and Satija (2018); Proserpio and Mahata (2016); Gaublomme et al. (2015); Avraham et al. (2015). This technology offers the potential to revolutionize infectious disease diagnostics by providing a more objective, molecular-level assessment of a patient's condition. For example, scRNA-seq has become a powerful tool in computational biology for analyzing complex genetic and molecular mechanisms at the resolution of individual cells Wang et al. (2023). By measuring the expression levels of each gene in every single cell within a sample, scRNA-seq provides detailed gene expression profiles, or transcriptomes, of a host. This comprehensive data effectively captures the cellular events occurring within the host, allowing for the study of complex coordinated responses, such as those of the immune system to infections. Importantly, different types of viral infections induce unique immune responses, which are reflected in the gene expression patterns of specific immune cells, as revealed by scRNA-seq Triana et al. (2021).

Despite the promise of scRNA-seq, several challenges have hindered its widespread adoption in clinical diagnostics. Firstly, the complexity of scRNA-seq data, which encompasses vast amounts of high-dimensional information, makes interpretation challenging without sophisticated analytical tools. Additionally, there is a lack of standardized analytical frameworks capable of effectively distinguishing between different infectious agents

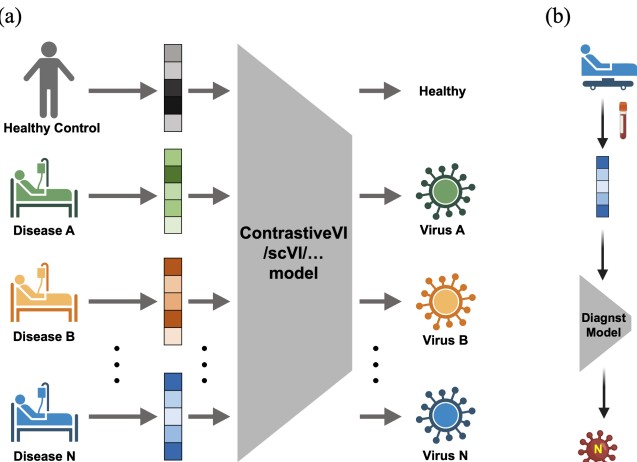

Figure 1: Schematic representation of the deep learning-based infectious disease diagnosis model. (a) Graphical overview of the training process for the deep learning model using patient-derived scRNA-seq data. (b) Graphical depiction of the application of the trained model in real-world clinical settings for diagnosing infectious diseases.

based on scRNA-seq profiles; also missing is a common data ground for comprehensively benchmarking any recent method. Moreover, the integration of scRNA-seq data from multiple sources introduces technical variations, known as batch effects, which can obscure biological signals Hicks et al. (2018); Tung et al. (2017). Furthermore, the potential of scRNA-seq for active pathogen identification in clinical settings remains largely unexplored.

To address these challenges and leverage the full potential of scRNA-seq in infectious disease diagnostics, we aim to develop a biological evidence-based diagnostic tool capable of identifying specific viral infections from patient blood samples with high accuracy and objectivity (as depicted in Figure 1 (b)), through combining advanced deep learning techniques with carefully curated scRNA-seq datasets. By analyzing the immune patterns from scRNA-seq profiles of infected patients' blood samples, we can establish a reference for each type of viral infection. Comparing a new patient's scRNA-seq profile against these established references allows us to accurately identify the viral pathogen causing the infection (as depicted in Figure 1 (a)).

Our approach involves several key components, beginning with meticulous data curation and preprocessing. We have sourced scRNA-seq datasets from patients with various viral infections, including COVID-19, influenza, and dengue. Our meticulous data preparation process involved rigorous filtering to include only high-quality cells, removing those with unusually high or low RNA counts, high mitochondrial gene expression, and excessive erythrocytic gene signatures. We applied normalization techniques, including count depth scaling and negative binomial regression, to correct for technical variations across cells. The resulting datasets comprised 20,351 high-quality cells for the Zhu dataset Zhu et al. (2020), 12,839 for the Waickman dataset Waickman et al. (2021), and 174,753 for the Blish dataset Wilk et al. (2020), representing the **largest and most comprehensive** of its kind to date. We provide visualizations to understand the underlying cellular heterogeneity.

To analyze this complex data, we evaluated the performance of latest deep learning models in distinguishing the unique gene expression patterns exhibited by scRNA-seq pro-

files from blood samples of patients infected with different viral pathogens. The models we examined include contrastiveVI Weinberger et al. (2023), scVILopez et al. (2018), SAVERHuang et al. (2018), scGPT,Cui et al. (2024) and PCA Baron et al. (2016) (as a sanity-check basline). These models were trained on scRNA-seq data obtained from patients at least one week post-infection. We then assessed the accuracy of these trained models in diagnosing the same diseases using scRNA-seq data, aiming to determine their effectiveness in identifying specific viral infections based on gene expression profiles. Our study involved a comprehensive evaluation of these models, assessing their efficacy in distinguishing scRNA-seq profiles associated with various viral infections, under a range of key quantitative and qualitative metrics. This approach aims to identify distinct immunological features representative of each viral infection, facilitating an evidence-based diagnostic method that improves accuracy and specificity in the infectious disease domain.

A significant challenge we have to address was the impact of batch effects when integrating datasets from various sources. These systematic, non-biological variations can obscure the biological variations of interest during data integration Benito et al. (2004); Johnson et al. (2007); Luo et al. (2010). To mitigate this issue, we employed both traditional and deep learning-based batch correction methods. We also conducted additional experiments using a different set of healthy controls collected by another group to assess whether diseased cell groups could still be distinguished from healthy cells collected from an independent source.

Our research consists of several significant contributions:

- **Novel Application Scenario**: This study is the first to leverage scRNA-seq data for the proactive identification of specific viral pathogens infecting a patient.

- **Data Curation and Preprocessing Framework**: We designed a tailored data curation and preprocessing framework to ensure consistency and minimize technical variation. This approach mitigated batch effects and preserved biologically meaningful signals, enabling robust evaluation of disease-specific immune responses.

- **Comprehensive Model Evaluation**: We evaluated various computational models, including contrastiveVI, scVI, SAVER, PCA, and scGPT, to assess their effectiveness, strengths, and limitations in distinguishing viral infections based on scRNA-seq profiles.

- **Addressing Batch Effects**: Our findings highlight the significant impact of batch effects in multi-source scRNA-seq data analysis, and the need for robust mitigations.

By addressing these aspects, our work not only advances the field of computational biology but also has potential implications for improving the accuracy and objectivity of infectious disease diagnostics. The integration of scRNA-seq data with advanced AI techniques represents a promising avenue for developing more precise and personalized diagnostic tools in clinical settings, potentially reducing diagnostic errors and improving patient outcomes. Our study demonstrates that deep learning models can accurately identify the type of infection from patient plasma samples based on scRNA-seq profiles, paving the way for more objective, evidence-based diagnostic methods in the infectious disease domain.

## 2 scRNA-seq for Viral Infections Diagnosis: Dataset Curation and Analysis

### 2.1 scRNA-seq for Infectious Disease Diagnosis

While scRNA-seq has revolutionized our understanding of cellular heterogeneity and gene expression dynamics Moignard et al. (2013, 2015); Nestorowa et al. (2016), its potential

for diagnosing infectious diseases remains largely unexplored. Previous studies have primarily focused on characterizing immune responses to diseases Li et al. (2022); Jagadeesh et al. (2022); Liu et al. (2024) but the application of scRNA-seq for proactive pathogen identification has been limited. Our work addresses this gap by providing **the first comprehensive dataset and benchmark study on scRNA-seq data for identifying viral pathogens in patient samples**.

The challenge in infectious disease diagnostics lies in developing a method that can accurately identify pathogens from a universal patient sample, moving beyond the current paradigm of symptom-based diagnosis followed by pathogen-specific tests. scRNA-seq offers a unique opportunity to capture the host's immune response at unprecedented resolution, potentially allowing for pathogen identification based on the distinct transcriptional signatures induced by different infections.

We aim to demonstrate that the unique patterns of immune cell activation, as revealed by scRNA-seq profiles of infected patients, can serve as robust identifiers of specific pathogens. To serve our goal, we will next describe our careful curation of diverse datasets, rigorous preprocessing pipeline, and comprehensive analysis of batch effects, that altogether provide a robust foundation for benchmarking computational models in the task of viral infection identification.

## 2.2 Multi-Source Data Collection

To ensure a robust evaluation, we carefully curated datasets representing three distinct viral infections: COVID-19, influenza, and dengue. Although both COVID-19 and flu belong to RNA viruses that cause respiratory tract infections, COVID-19 exhibits distinct clinical characteristics from flu, such as higher morbidity and mortality rates. By including dengue virus infections alongside respiratory viruses, we aimed to test the method's ability to distinguish between infections with potentially similar and disparate clinical presentations. Our data collection strategy was designed to address **two key challenges: minimizing batch effects and representing diverse pathogens**.

Our dataset comprises scRNA-seq data from multiple sources. The Zhu dataset provided 20,351 high-quality cells, including samples from three healthy individuals, five COVID-19 patients, and two influenza A virus (IAV)-infected patients. The COVID-19 patients were within five to ten days post symptom onset and confirmed positive through nucleic acid testing. For dengue infections, we utilized the Waickman dataset, which contributed 12,839 cells from four dengue patients. This data was generated using the 10x Genomics 5' capture gene expression platform, with an average sequencing depth of 100,000 reads per cell and approximately 5,700 cells per library. To enhance our analysis of healthy controls, we further incorporated the Blish dataset, which added 174,753 cells from 41 individuals (8 healthy and 33 COVID-19 patients). Table 1 overviews the dataset statistics.

Note the Zhu dataset is a strategic choice, which includes both COVID-19 and influenza data collected simultaneously by the same research group. This choice helps isolate disease-specific signals from technical variations ("batch effect"), which will be discussed more later.

Further details regarding the datasets used in this study, including dataset links and additional supporting information, can be found at this page Supporting Information.

## 2.3 Preprocessing and Quality Control

Our preprocessing pipeline was designed to address the unique challenges arising from working with multi-source scRNA-seq data for diagnostic purposes. We implemented rigorous filtering criteria to ensure high-quality data representative of the immune response.

Table 1: Dataset Statistics.

| Dataset | Disease | Individual | # cells |
|---|---|---|---|
| Blish Wilk et al. (2020) | COVID | 8 Healthy + 33 COVID | 174,753 |
| Zhu Zhu et al. (2020) | COVID + Flu | 3 Healthy + 4 COVID + 2 Flu | 20,351 |
| Waickman Waickman et al. (2021) | Dengue | 4 Dengue | 12,839 |

Cells with unusually high or low total RNA counts, high mitochondrial gene expression (indicative of cell stress or apoptosis), and other outliers were excluded. Specifically, we retained only cells with 300-6,500 unique features and less than 10% mitochondrial RNA content. To eliminate erythrocyte contamination, cells with more than 5% erythrocytic gene signatures (defined by the expression of HBB, HBA1, and HBA2) were also excluded.

To enable meaningful comparisons across samples and datasets, we applied normalization techniques including count depth scaling and negative binomial regression for multi-sample integration. This process corrects for differences in sequencing depth and other technical variations across cells. Following these preprocessing steps, our final datasets comprised 20,351 high-quality cells from the Zhu dataset, 12,839 from the Waickman dataset, and 174,753 from the Blish dataset: Table 1 provides an overview of our curated data.

Further details regarding the preprocessing and quality control of data used in this study can be found at this page Supporting Information.

## 2.4 Data Visualization and Analysis

To visualize and understand the underlying cellular heterogeneity in our data, we employed Uniform Manifold Approximation and Projection (UMAP).UMAP transforms high-dimensional gene expression data into a lower-dimensional space McInnes et al. (1802); Yang et al. (2021) allowing us to effectively visualize distinct cell clusters while preserving both local and global structures within the data.

Through integrated analysis of the combined Zhu and Waickman datasets, we identified 15 statistically significant populations corresponding to all major leukocyte subsets, including monocytes and lymphocytes. This comprehensive representation of diverse immune cell types, as shown in Figure S1, ensures that our subsequent analyses capture a wide range of immune response signals, thus providing a strong foundation for our benchmarking studies.

**Addressing Batch Effects** While batch effects are a well-known challenge in scRNA-seq analysis, their impact on diagnostic applications remains understudied. Figure S2 demonstrated such systematic, non-biological variations in our curated multi-source dataset, that can obscure true biological differences during data integration.

Our work explicitly addresses this gap through strategic dataset selection and comparative experimental analysis (in Section 4) of batch correction methods. By using the Zhu dataset for both COVID-19 and influenza, we minimized batch effects between these two infections, allowing for a more direct comparison of their transcriptional signatures. To investigate the impact of batch effects, we performed additional experiments using the Blish dataset as an independent source of healthy controls. This allowed us to assess whether diseased cell groups could still be distinguished from healthy cells collected under different technical conditions. We further explored various batch correction methods, including both traditional approaches and deep learning-based techniques.

## 3 Applying scRNA-seq Computational Models to Proactive Pathogen Identification for Viral Infections Diagnosis

While there have been efforts to analyze scRNA-seq data for single infections or single cell types, integrating multiple infection types (e.g., COVID-19, dengue, and influenza) and multiple cell types (e.g., various immune cells in PBMCs) within the same dataset remains unexplored. Our work addresses this gap by presenting the first comprehensive approach for disease diagnosis that leverages a holistic understanding of the immune response across different infections and cell types.

We recognized the challenge of selecting the best model that can comprehensively understand the unique and potentially subtle transcriptional signatures arising from infections. The differences between cells are not only due to the infecting pathogen but also intrinsic factors such as cell type diversity and data source variability. Identifying transcriptional signatures that exclusively arise from unique viral infections is critical. Therefore, we focused on evaluating each model's capability to distinguish cells based solely on infection-induced gene expression differences.

Our contribution lies not in the development of these models themselves, but in the novel application and comprehensive evaluation of these models for infectious disease diagnosis. We aim to demonstrate, for the first time, the strengths and limitations of each model in distinguishing transcriptional signatures between differentially infected cells and healthy cells using scRNA-seq data. We evaluated and compared their performance based on key metrics: adjusted random index (ARI) score, normalized mutual information (NMI) score, and silhouette score (S-score).

Next, we describe each model we benchmarked and how they are applied in our study. All models except SAVER extract feature representations for the sRNA-seq data.

**Principal Component Analysis (PCA):** We utilized PCA as a baseline to check for sanity. Our approach involved entering a normalized RNA count matrix into PCA, where rows represented genes, columns represented cells, and each entry denoted the expression level of a gene in a specific cell. By outputting a set of principal components (which are essentially linear combinations of the original gene expression data), we can capture the most significant variations within the dataset.

**ContrastiveVI:** By leveraging variational autoencoder (VAE) and contrastive learning, contrastiveVI accounts for uncertainties in observed RNA counts and integrates shared and treatment-specific factors Weinberger et al. (2023) in distinguishing the unique transcriptional profiles of infected patients. We train the model on our curated data, with an RNA count matrix and labels indicating cell origins (background or target dataset). contrastiveVI robustly differentiates intricate gene expression patterns associated with various viral infections.

**Single-cell Analysis Via Expression Recovery (SAVER):** We also leveraged the SAVER model to recover true gene expression levels in each cell, effectively removing technical variation while preserving biological variation Huang et al. (2018). SAVER models the count of each gene in each cell using a Poisson-Gamma mixture. It estimates gamma prior parameters using an empirical Bayes-like approach, utilizing Poisson-Lasso regression with the expression levels of other genes as predictors. After that, SAVER outputs the posterior distribution of the true expression, quantifying uncertainty. We used the posterior mean as the recovered gene expression value.

**scGPT:** We also explored scGPT, a generative pre-trained transformer model specifically designed for scRNA-seq data analysis Cui et al. (2024). This model utilizes stacked transformer layers with multi-head attention to learn embeddings for both cells and genes simultaneously, and its input is an RNA count matrix derived from scRNA-seq data. scGPT was pre-trained on extensive scRNA-seq datasets, such as those from the CellXGene portal,

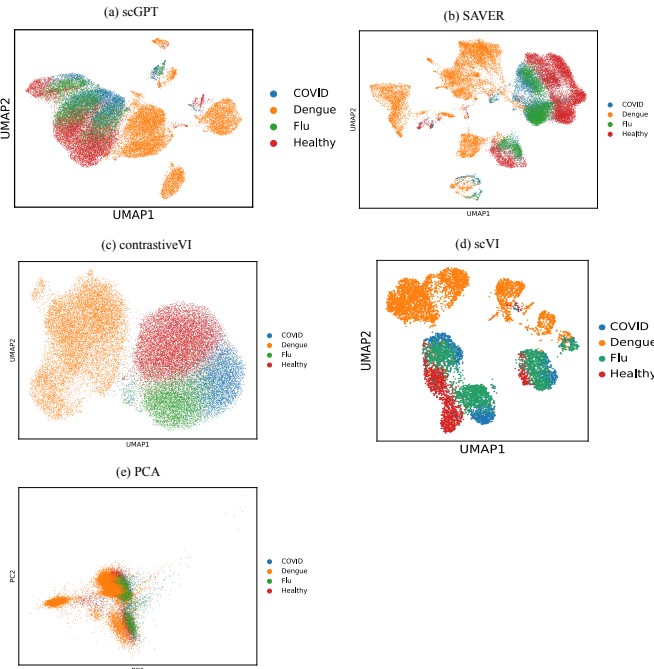

Figure 2: The UMAP visualizations of clustering outcomes, as demonstrated by various models indicated above each plot, are presented. These models were trained on the Zhu dataset (control, COVID-19, and flu) and the Waickman dataset (dengue). No batch correction was performed.

using a specially designed attention mask and a generative training pipeline that optimizes cell and gene representations in a self-supervised manner. Beyond assessing scGPT's capability in distinguishing differentially infected cells, we also test the model's built-in ability for batch correction by directly processing our data through the model.

**Single-Cell Variational Inference (scVI):** scVI offers a scalable framework for the probabilistic representation and analysis of single-cell gene expression data Lopez et al. (2018). We leveraged its hierarchical Bayesian structure, where conditional distributions are defined by deep neural networks, to gain insights into complex transcriptional landscapes. By encoding each cell's transcriptome through a nonlinear transformation, scVI transforms the data into a low-dimensional latent vector composed of normal random variables. This latent representation is then decoded via another nonlinear transformation to generate posterior estimates of distributional parameters for each gene in every cell. By assuming a zero-inflated negative binomial distribution, scVI effectively accommodates the observed over-dispersion and limited sensitivity inherent in scRNA-seq data, allowing us to robustly model and interpret gene expression patterns across diverse cell types and infection statuses.

## 4 Result and Analysis

### 4.1 Evaluation Metrics

**Quantitative Metrics:** We calculated several metrics, including ARI, NMI, S-score, to systematically evaluate the performance of each model in clustering cells in UMAP by disease type. **ARI** measured the similarity between clustering results and ground truth classifications, adjusting for chance groupings, which provides a more accurate assessment of clustering quality. **NMI** was used to evaluate the amount of information shared between the predicted and true clusterings, normalized for cluster size, ensuring the metric remains informative regardless of the number of clusters. **S-score** was employed to assess both the cohesion within clusters and the separation between clusters, offering insights into the compactness and distinctness of the clusters formed. These metrics were chosen because they are widely used in computational biology to evaluate clustering in single-cell RNA-seq (scRNA-seq) data, where capturing biologically meaningful cell groupings is more relevant than classification accuracy alone. Unlike traditional metrics such as AUC, precision, and recall, which are designed for discrete classification tasks, ARI, NMI, and S-score account for the high-dimensional, heterogeneous nature of scRNA-seq data. By leveraging these domain-specific metrics, we ensured that our evaluation focused on preserving biologically relevant transcriptional patterns, providing a more meaningful assessment of model performance.

We also apply $k$**-nearest neighbors** onto the extracted feature representations (except SAVER) to classify the test data by identifying the nearest labeled training points, and report accuracies with $k = 1$ and 5.

**Qualitative Metrics:** While the above quantitative metrics remain indispensable for assessing model performance, the visual clustering performance on UMAP plays a vital role in our benchmarking. Practically, this approach enhances our confidence in the practical applicability of the models and their potential for translation into clinical settings, by ensuring the biological relevance and interpretability of our deep learning-based infectious disease diagnostic models.

Visual representation provides an intuitive understanding of model behavior, offering a clear illustration of how effectively the model differentiates between disease types. This approach is particularly valuable as it can reveal nuances that may not be captured by numerical metrics alone. Although a model might achieve high scores on quantitative measures such as ARI, NMI, and S-score, visual inspection can reveal whether the cluster-

ing incorporates cell-specific features that may be irrelevant to our diagnostic goals. By examining UMAP plots, we can verify that the observed clustering patterns align with biologically meaningful distinctions between diseases, rather than reflecting arbitrary or irrelevant cellular characteristics.

This focus on visual representation serves as a crucial additional layer of validation. It allows us to confirm that the clustering patterns reflect features that are not only statistically significant but also biologically relevant and practically useful for disease identification. Such interpretability is essential for diagnosing potential shortcomings in model performance that may not be apparent from numerical metrics alone – whether the model is focusing on irrelevant features or failing to distinguish meaningful patterns altogether.

Table 2: Clustering performance on Zhu Zhu et al. (2020) and Waickman Waickman et al. (2021) datasets. Note: SAVER model nearest-neighbor accuracy is not included as it does not produce a latent space representation.

| Training Dataset | Method | ARI | NMI | S-score | Accuracy (K=1) | Accuracy (K=5) |
|---|---|---|---|---|---|---|
| | contrastiveVI | 0.3871 | 0.5597 | 0.3594 | 0.88 | 0.90 |
| | scVI | 0.3822 | 0.5574 | 0.3594 | 0.85 | 0.86 |
| Zhu + Waickman | SAVER | 0.1884 | 0.4245 | 0.0973 | N/A | N/A |
| | PCA | 0.2374 | 0.3985 | -0.0744 | 0.83 | 0.87 |
| | scGPT | 0.1982 | 0.3789 | 0.1224 | 0.86 | 0.88 |

## 4.2 Performance Comparison and Analysis

Our comparative analysis firstly revealed that contrastiveVI and scVI demonstrated the strongest performance in cell clustering *without batch correction* (see Table 2 and Figures 2 (c) & (d)). More specifically:

- The UMAP plot highlights that contrastiveVI generates the most distinct clustering pattern, correlating with each infection type. This clear separation of diverse immune cells based on infection type showcases contrastiveVI's capability to identify unique transcriptional signatures associated with specific viral infections. We further validated this finding by incorporating a different set of healthy controls collected by another group (Blish et al., see Table 1). The disease-specific clustering remained distinct from the healthy cells, even when using an independent source (see Figure S3). This robustness and reliability of contrastiveVI approach reinforce its potential as a universal diagnostic tool.

- While scVI delivered strong quantitative metrics, its ability to visually differentiate between COVID-19 and influenza in the UMAP plot was limited, as evident in Figure 2 (d). The disparity between scVI's quantitative performance and its UMAP visualization can be attributed to its intrinsic normalization and batch correction mechanisms. While these built-in processes effectively optimize clustering metrics, they appear to have unexpected consequences on the UMAP representation. UMAP's sensitivity to subtle data transformations may amplify these corrections, resulting in less distinct visual separation between the two viral infections.

  Moreover, scVI's performance suggests an influence of cell type-specific information, evidenced by the Dengue cluster's separation into three distinct groups. This indicates a potential bias towards cell type features, implying that while scVI excelled in quantitative measures, it may have been influenced by information not directly relevant to our primary goal of disease differentiation.

The identification of these nuances through UMAP visualization highlights the critical importance of combining quantitative metrics with qualitative visual analysis in evaluating model performance for high-dimensional data tasks. This comprehensive approach provides a more holistic understanding of model behavior, revealing subtleties that may be obscured by numerical metrics alone. Such insights are crucial for developing robust and clinically relevant diagnostic tools based on scRNA-seq data.

- SAVER, like scVI, struggled to clearly differentiate between COVID-19 and influenza in the UMAP visualization. This challenge may stem from the inherent similarities in immune responses to these respiratory tract infections, making their transcriptional signatures difficult to distinguish. Furthermore, the SAVER-generated UMAP showed the dengue population dispersed across multiple clusters rather than forming a cohesive group. This dispersion likely results from SAVER's underlying statistical assumptions about data distribution, which may not accurately reflect the complex reality of viral infection transcriptomics. The model's inability to capture infection-specific transcriptional signatures effectively could be attributed to the mismatch between its assumed data distribution and the actual distribution of the scRNA-seq profiles. This discrepancy appears to have limited SAVER's capacity to accurately map the multidimensional relationships between different viral infections in the lower-dimensional UMAP space.

- scGPT exhibited an intriguing behavior, clustering cells predominantly based on their cellular identity rather than their infection status. This outcome can be attributed to scGPT's pre-training process, which heavily emphasized cell type-specific transcriptional patterns rather than disease-associated signatures. As a result, when applied to our dataset, scGPT prioritized the identification of cell type-associated transcriptional features over those indicative of different infections. This preference for cell type classification demonstrates both the power and the limitation of transfer learning in this context. While scGPT's ability to discern cell types even in a disease-focused dataset is impressive, it also reveals a crucial limitation in its current form for infection-specific diagnostics. This finding underscores the importance of tailoring pre-training strategies and fine-tuning processes to the specific task at hand, particularly when repurposing general-purpose models for specialized biomedical applications like infectious disease diagnosis.

- Lastly, PCA demonstrated the least effective performance, struggling to make clear distinctions between various disease states and healthy controls. The complex, non-linear relationships that characterize immunological responses to different pathogens are not adequately captured by PCA's straightforward linear transformations, underscoring the necessity for more sophisticated, non-linear approaches.

Overall, our experiments have demonstrated the significant potential of deep learning models in leveraging patient-derived scRNA-seq data for infectious disease diagnostics. While the visual representations in UMAP plots varied in their clarity of disease-specific clustering across different models, a remarkable consistency emerged in their quantitative performance. Each model achieved an accuracy of approximately 85% in distinguishing between different infectious states (as shown in Table 2). This high level of accuracy, maintained across diverse computational approaches, strongly validates the robustness and reliability of our scRNA-seq-based diagnostic strategy.

## 4.3 Mitigating the Cell Types Bias: Performance on T cell-only Data

To address models' potential bias of clustering based on cell types rather than disease-specific variations, we create a subset by focusing exclusively on T cell data from the blood samples. We aimed to minimize cell type-specific variations, hypothesizing that this would allow the models to better capture disease-specific transcriptional signatures.

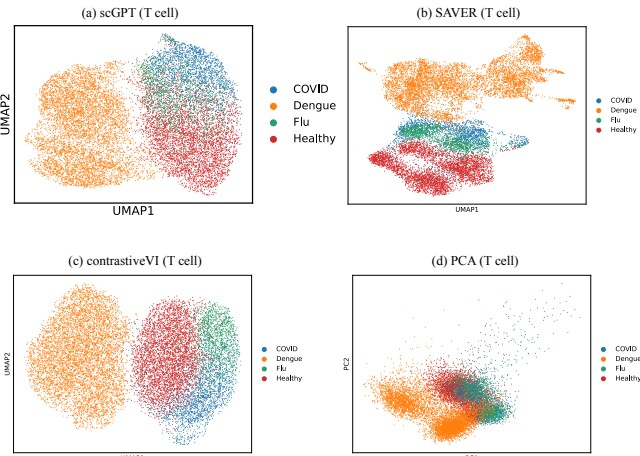

Figure 3: The UMAP visualization of clustering outcomes, as demonstrated by various models indicated in the figure above each plot, is presented. These models were trained exclusively on scRNA-seq data from T cells in patient blood samples, using the Zhu (control, COVID-19, and flu) and Waickman (dengue) datasets. No batch correction was performed.

This approach yielded intriguing results, especially for scGPT. When trained solely on T cell data, scGPT generated a more coherent dengue cluster in the UMAP visualization (Figure 3 (a)). This contrasts sharply with its performance on the complete PBMC dataset (Figure 2 (a)), where it separated dengue data into five distinct clusters, indicative of its cell type-specific bias. Similarly, SAVER and PCA (Figures 2 (b) & (e) and Figures 3 (b) & (d)), when trained on T cell data, observed a similar shift in the dengue clusters to a less separated state compared to their PBMC-trained counterparts. This shift demonstrates that limiting the dataset to a single cell type can indeed help models like scGPT, SAVER, and PCA focus more on disease-specific variations, although it did not fully achieve distinct disease-specific clustering across all conditions.

However, the impact of this strategy was not uniformly positive across all models. Notably, contrastiveVI's performance showed a slight decline when trained on T cell-only data (Figure 3 (c)), as evidenced by increased overlap between disease-specific clusters in the UMAP visualization. This observation suggests that the diverse immune cell population present during an infection contributes to a more robust and distinctive transcriptional signature, which aids in accurately identifying the infection type. These findings highlight a crucial trade-off in model development for infectious disease diagnostics. While reducing cell type-specific variations can help focus on disease-specific patterns, it may also diminish the overall richness and robustness of the transcriptional signature. The diverse immune cell response appears to provide important contextual information that enhances the model's ability to distinguish between different infections.

Our investigation into the impact of cell type diversity on model performance underscores the complexity of developing effective diagnostic tools using scRNA-seq data. It emphasizes the need for a nuanced approach in data selection and preprocessing, balancing cell type diversity and disease-specific variations, to develop more robust, comprehensive, and well-balanced deep models.

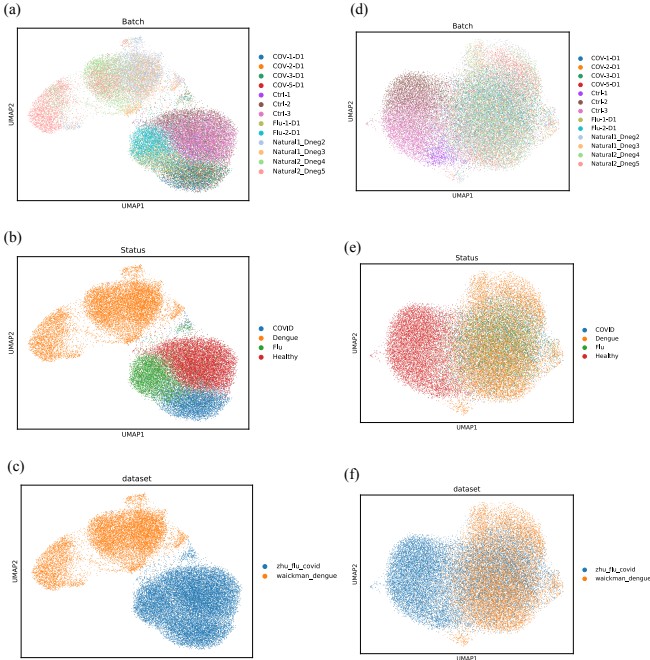

Figure 4: The UMAP visualization of the clustering outcomes demonstrated by contrastiveVI is presented. The model was trained on the Zhu (control, COVID-19, and flu) and Waickman (dengue) datasets. Panels (a)-(c) show the results after batch correction using the built-in function of scGPT prior to training. Each point on the UMAP represents an individual cell, color-coded based on its (a) batch, (b) type of infection, or (c) dataset of origin. Panels (d)-(e) display the results after batch correction using the built-in function of contrastiveVI prior to training. Each point on the UMAP represents an individual cell, color-coded based on its (d) batch, (e) type of infection, or (f) dataset of origin.

### 4.4 Attempts to Mitigate the Batch Effect

The integration of datasets from multiple sources introduces significant challenges in the form of batch effects, which can obscure true biological signals. As illustrated in Figure S3, while S3 (a) shows cells occupying distinct, disease-specific regions in the UMAP space when color-coded by disease type, S3 (b) reveals clustering predominantly based on dataset origin. This stark contrast suggests that observed transcriptional differences may be largely attributable to technical variations in sample collection or processing methods rather than genuine disease-specific features.

To address this issue, we explored various batch correction strategies. Initially, we applied the built-in batch correction function of contrastiveVI to the combined Zhu and Waickman datasets. However, this approach resulted in a complete loss of distinction between disease groups (Figure 4 (b)). This outcome likely stems from the algorithm's indiscriminate reduction of overall variance, inadvertently normalizing disease-specific transcriptional features alongside batch effects (Figures 4 (a) and 4 (c) ). The challenge lies in the algorithm's inability to distinguish biologically relevant variations from technical noise without prior knowledge, potentially eliminating crucial disease-specific signals in the

process. We further investigated batch correction using scGPT prior to training with contrastiveVI. While this method preserved more of the disease-specific features compared to contrastiveVI's built-in correction (Figure 4 (e)), it still fell short in effectively eliminating batch effects, as evidenced by persistent batch-specific (Figure 4 (d)) and dataset-specific (Figure 4 (f)) clustering in the UMAP visualization.

Despite these challenges, we validated our overall approach through strategic dataset selection. We minimized intrinsic batch effects by utilizing the Zhu dataset, which contains scRNA-seq data for both COVID-19 and influenza collected by a single research group. This careful selection allowed us to demonstrate that our benchmarked models can generate distinct clustering patterns correlating with different infection types and healthy controls, even without perfect batch correction. Our experience highlights a pragmatic approach to developing deep learning-based diagnostic tools for infectious diseases using scRNA-seq data. In the current absence of universally effective batch correction methods, focusing on single-source, multi-disease datasets can ensure the integrity of the diagnostic process. This strategy allows for meaningful progress in the field while emphasizing the need for more advanced batch correction techniques in future research.

## 5 Conlcusion

This study presents a pioneering exploration of deep learning models for distinguishing between different viral infections and healthy controls using scRNA-seq profiles. Our work not only demonstrates the feasibility of using scRNA-seq data for infectious disease diagnosis but also provides a comprehensive benchmark for future studies in this field. By leveraging the power of deep learning and high-dimensional genomic data, we pave the way for more precise, efficient, and personalized approaches to infectious disease management.

As we continue to refine these methods and address the challenges identified, the integration of scRNA-seq-based diagnostics into clinical practice holds the promise of transforming our ability to rapidly and accurately identify viral pathogens, ultimately improving patient outcomes and public health responses to infectious diseases. Looking ahead, several avenues for future research emerge from our findings. First, expanding the range of pathogens included in the training data could further validate the generalizability of our approach. Second, investigating the integration of other omics data, such as proteomics or metabolomics, could provide complementary information to enhance diagnostic accuracy. Finally, clinical validation studies will be crucial to translate these computational findings into practical diagnostic tools.

## 6 Competing interests

No competing interest is declared.

## 7 Author contributions statement

Z.Y. and X.C. conceived the experiment(s), Z.Y., X.C. and B.Z. conducted the experiment(s). Z.Y. , X.C. and T.C. analyzed the results. Z.Y. and Z.W. wrote and reviewed the manuscript.

## 8 Acknowledgments

This work is supported in part by funds from the NSF AI Institute for Foundations of Machine Learning (IFML).

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

**Supplementary Information**

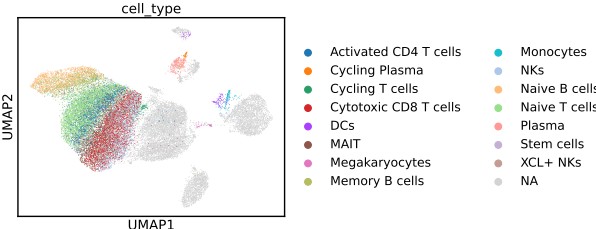

Figure S1: UMAP visualization of clustering outcomes generated by scGPT, trained on the Zhu dataset (control, COVID-19, and flu) and the Waickman dataset (dengue), comprising a total of 33,190 cells from 13 donors. Each point on the UMAP represents an individual cell, color-coded based on its cell type. No batch correction was performed.

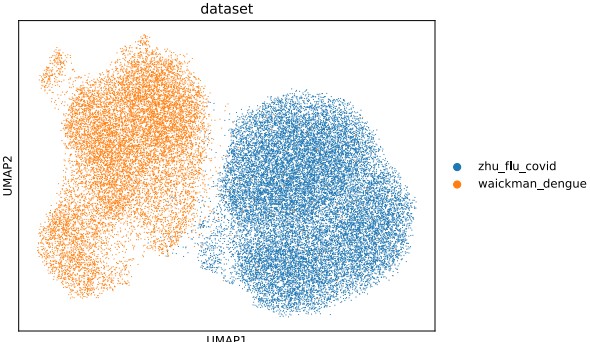

Figure S2: UMAP visualization of clustering outcomes generated by contrastiveVI, trained on the Zhu dataset (control, COVID-19, and flu) and the Waickman dataset (dengue). Each point on the UMAP represents an individual cell, color-coded based on its dataset of origin. No batch correction was performed.

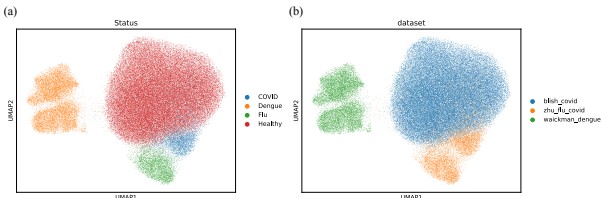

Figure S3: UMAP visualization of clustering outcomes generated by contrastiveVI, trained on the Zhu dataset (COVID-19 & flu), the Waickman dataset (dengue), and the Blish dataset (control). No batch correction was performed. (a) Each point on the UMAP represents an individual cell, color-coded based on its type of infection. (a) Each point on the UMAP represents an individual cell, color-coded based on its dataset of origin.

