# OpenReview forum: "Deep Learning for Accurate Diagnosis of Viral Infections through scRNA-seq Analysis: A Comprehensive Benchmark Study"
_DMLR — Accepted by DMLR_

### Review · Reviewer_kmX3 · 2024-11-18

**Recommendation:** 4
**Confidence:** 3

**Summary Of Contributions:**

This paper presents a comprehensive benchmark study on the use of deep learning models to accurately diagnose viral infections from patient-derived single-cell RNA sequencing (scRNA-seq) profiles. The researchers curated a large and diverse dataset of scRNA-seq data from patients infected with COVID-19, influenza, and dengue, addressing challenges such as batch effects and data heterogeneity. They evaluated the performance of several deep learning models, including contrastiveVI, scVI, SAVER, and scGPT, in distinguishing the unique gene expression patterns associated with different viral infections. The results demonstrate that the contrastiveVI model outperforms other methods in accurately identifying the type of viral infection based on the scRNA-seq profiles. The study highlights the substantial impact of batch effects when analyzing scRNA-seq data from multiple sources and the need for robust mitigation strategies. Overall, this research contributes to the development of more objective, evidence-based diagnostic methods in the infectious disease domain, potentially reducing diagnostic errors and improving patient outcomes.

Pros

The study tries to leverage scRNA-seq data for the proactive identification of specific viral pathogens infecting a patient is novel.
The researchers established a comprehensive methodology for preparing and integrating scRNA-seq datasets from multiple sources, addressing challenges such as batch effects and data heterogeneity.
The study comprehensively evaluated various computational models, including contrastiveVI, scVI, SAVER, PCA, and scGPT, to assess their effectiveness in distinguishing viral infections based on scRNA-seq profiles.
The findings highlight the significant impact of batch effects in multi-source scRNA-seq data analysis and the need for robust mitigation strategies.
The integration of scRNA-seq data with advanced AI techniques represents a promising avenue for developing more precise and personalized diagnostic tools in clinical settings, potentially reducing diagnostic errors and improving patient outcomes.
Cons

It seems that the study focuses solely on covid & flu and does not consider the potential application of the proposed approach to other types of infectious diseases or non-infectious conditions.
It seems that the study does not include an external validation dataset to assess the generalizability of the proposed models to new, unseen patient populations.
Overall, this paper presents a well-designed and comprehensive study that demonstrates the potential of deep learning-based approaches for accurate viral infection diagnosis using scRNA-seq data.

**Strengths:**

none

**Audience:**

Yes

**Claims And Evidence:**

none

**Datasets And Benchmarks:**

none

**Extended Submissions:**

none

**Requested Changes:**

none

**Strengths And Weaknesses:**

none

---

### Review · Reviewer_aii3 · 2024-11-25

**Recommendation:** 3
**Confidence:** 2

**Summary Of Contributions:**

The authors focus on scRNA-seq data and establish a robust framework for curating and integrating multi-source scRNA-seq datasets. The paper addresses challenges like batch effects and data heterogeneity, and various computational models are evaluated for their effectiveness in detecting viral infections.

**Strengths:**

Please see the previous sections for strengths.

**Audience:**

Yes

**Claims And Evidence:**

n/a

**Datasets And Benchmarks:**

An URL link is not available in the current script.

**Extended Submissions:**

n/a

**Limitations:**

Please see the previous section on pros&cons.

**Requested Changes:**

Please see the comment on evaluation metrics.

**Strengths And Weaknesses:**

Strengths:
1. The application of scRNA-seq data is interesting and meaningful. It helps make AI applied in more beneficial areas.
2. The paper provides a clear illustration on data collection and the evaluation results of multiple methods. Such an illustration show the data is able for benchmarking.
3. The paper also provides interesting discussion on the future direction of the paper.

Weaknesses:
1. The paper didn't explain why certain metrics are not used for evaluation. For classification tasks, the common one would be like AUC, Precision, recall and so on. Maybe this is a specific domain needs other metrics. But it is not explained here.
2. An URL link is not available in the current script.

---

### Review · Reviewer_yQn8 · 2025-01-05

**Recommendation:** 3
**Confidence:** 3

**Summary Of Contributions:**

1. This study aims to leverage scRNA-seq data for the proactive identification of specific viral pathogens infecting a patient. To that end, authors implemented contrastiveVI, scVI, SAVER, PCA, and scGPT to assess their effectiveness, strengths, and limitations in distinguishing viral infections based on scRNA-seq profiles.
2. This paper investigated challenges for integrating scRNA-seq datasets from multiple sources such as batch effects and data heterogeneity.

**Strengths:**

The paper is well-organized, and the submission seems appropriate for DMLR. The dataset constructed is of broad interest in benchmarking and facilitating research on AIML methods for scRNA-seq Analysis, especially for diagnosis of viral infections. The research is of good quality, challenges in the analysis is well-discussed.

**Audience:**

Yes

**Broader Impact Concerns:**

no concerns

**Claims And Evidence:**

Yes, the claims made in the submission are supported by convincing evidence

**Datasets And Benchmarks:**

Some details on dataset are disclosed, but the url to the dataset is not available in the submission

**Extended Submissions:**

submission is not an extended version of a previously published work

**Limitations:**

The weakness and limitations are summarized in the above section. Overall, I think this is a good paper with solid contributions.

**Requested Changes:**

1. The dataset itself is not open-access, there is no url to dataset itself provided. More details on preprocessing details should be disclosed. The url of this dataset should be provided for proper evaluations to be made.

2. More discussion on prior works on methodologies of addressing batch effects should be enhanced, if the authors want to claim novelty in this perspective.

3. Technical details on preprocessing should be further discussed:
a). The authors claim they implemented rigorous filtering criteria to ensure high-quality data representative of the immune response. "Cells with unusually high or low total RNA counts, high mitochondrial gene expression (indicative of cell stress or apoptosis), and other outliers were excluded. Specifically, we retained only cells with 300-6,500 unique features and less than 10% mitochondrial RNA
content."
How were the cutoff values determined? From data driven approaches or empirical evidence? Are these numbers applicable for other types of scRNA-seq analyses

b). The authors adopted normalization techniques including count depth scaling and negative binomial regression for multi-
sample integration to enable meaningful comparisons across samples and datasets.
I understand the purpose of applying count depth scaling. However, it is not clear why negative binomial regression (which is commonly used for modeling over-dispersed count outcome variables) was adopted. More discussions regarding this should be made, and please consider adding some notations and equations for improving clarity.

4. The authors used visual clustering performance on UMAP as qualitative metric, and claims "visual inspection can reveal whether the clustering incorporates cell-specific features that may be irrelevant to our diagnostic goals. By examining UMAP plots, we can verify that the observed clustering patterns align with biologically meaningful distinctions between diseases, rather than reflecting arbitrary or
irrelevant cellular characteristics."

It is not clear why latent space learned by UMAP can help to achieve such aforementioned goals, especially considering the UMAP components/spaces may not be directly comparable for each experiment. Please include more details on how to attribute UMAP components to features.

**Strengths And Weaknesses:**

Strengths:
1. This study aims to leverage scRNA-seq data for the proactive identification of specific viral pathogens infecting a patient. To that end, authors implemented contrastiveVI, scVI, SAVER, PCA, and scGPT to assess their effectiveness, strengths, and limitations in distinguishing viral infections based on scRNA-seq profiles.
a). The dataset used for investigation comprises scRNA-seq data from multiple sources, including different types of infections, e.g., influenza A virus, COVID-19 and dengue, as well as healthy individuals for control groups.
b). The authors adopted recently proposed scRNA-seq computational models to proactive pathogen identification for viral infections diagnosis

2. The authors paid special attention to challenges such as batch effect, and investigated several approach for addressing this issue.
a). The author selected the Zhu dataset, which includes both COVID-19 and influenza data collected simultaneously by the same research group, for addressing batch effect.
b). To investigate the impact of batch effects, the authors performed additional experiments using the Blish dataset as an independent source of healthy controls.

Weakness:
1. The dataset itself is not open-access, there is no url to dataset itself provided. More details on preprocessing details should be disclosed.

2. The author claims they proposed a methodology for addressing batch effect, however, it is somewhat ad-hoc not a methodology that can be adopted for analyses of other purpose of scRNA-seq.

More discussion on prior works on methodologies of addressing batch effects should be enhanced, if the authors want to claim novelty in this perspective. In this case, they are able to adopt a dataset which  includes both COVID-19 and influenza data collected simultaneously by the same research group. However, there is no guarantee that such types of datasets also exist for other applications of scRNA-seq. Therefore, I think referring the approach as a methodology seems a little inappropriate, especially considering there are many works available on addressing batch effects.

3. Technical details on preprocessing should be further discussed:
a). The authors claim they implemented rigorous filtering criteria to ensure high-quality data representative of the immune response. "Cells with unusually high or low total RNA counts, high mitochondrial gene expression (indicative of cell stress or apoptosis), and other outliers were excluded. Specifically, we retained only cells with 300-6,500 unique features and less than 10% mitochondrial RNA
content."

How were the cutoff values determined? From data driven approaches or empirical evidence? Are these numbers applicable for other types of scRNA-seq analyses

b). The authors adopted normalization techniques including count depth scaling and negative binomial regression for multi-
sample integration to enable meaningful comparisons across samples and datasets.
I understand the purpose of applying count depth scaling. However, it is not clear why negative binomial regression (which is commonly used for modeling over-dispersed count outcome variables) was adopted. More discussions regarding this should be made, and please consider adding some notations and equations for improving clarity.

4. The authors used visual clustering performance on UMAP as qualitative metric, and claims "visual inspection can reveal whether the clustering incorporates cell-specific features that may be irrelevant to our diagnostic goals. By examining UMAP plots, we can verify that the observed clustering patterns align with biologically meaningful distinctions between diseases, rather than reflecting arbitrary or
irrelevant cellular characteristics."

It is not clear why latent space learned by UMAP can help to achieve such aforementioned goals, especially considering the UMAP components/spaces may not be directly comparable for each experiment. Please include more details on how to attribute UMAP components to features.